# Prevalence of vision impairment among patients with diabetes mellitus in sub-Saharan Africa: A systematic review and meta-analysis

Altaseb Beyene Kassaw[1]*, Addis Alem Hadigu[1], Melese Shenkut Abebe[1], Amare Abera Tareke[1], Wondwossen Debebe[1], Gosa Mankelkl[1], Natnael Kebede[2], Alemu Gedefie[3]

1 Department of Biomedical Science, College of Medicine and Health Science, Wollo University, Dessie, Ethiopia, 2 School of Public Health, College of Medicine and Health Science, Wollo University, Dessie, Ethiopia, 3 Department of Medical Laboratory Science, College of Medicine and Health Science, Wollo University, Dessie, Ethiopia

* altasebbeyene7@gmail.com

## Abstract

### Background

Diabetes mellitus is a growing global public health challenge, especially in low- and middle-income regions like sub-Saharan Africa, where healthcare resources are often limited. One of its most debilitating complications is vision impairment, which significantly impacts the quality of life, productivity, and independence of individuals. Although several studies have assessed the prevalence of vision impairment among individuals with diabetes in sub-Saharan Africa, their findings have been inconsistent and fragmented across different areas. A comprehensive synthesis of this evidence is crucial for informing healthcare planning, early screening, and management strategies. This study aimed to systematically review and synthesize the existing evidence on the prevalence of vision impairment among patients with diabetes mellitus in the region.

### Methods

A literature exploration was done in an electronic database such as PubMed, Google Scholar, Web of Science, Hinary, and African Journals Online. We included all observational studies reporting the prevalence of vision loss among diabetes mellitus patients in Sub-Saharan Africa. A random-effect meta-analysis model was computed to estimate the pooled prevalence of vision impairment. $I^2$ test and Cochrane Q test statistics were used to assess the heterogeneity of the studies. Subgroup analysis was done based on the countries where the research was conducted. STATA Version 16 statistical software was used for data analysis. Publication bias was examined by funnel plots and Egger's tests.

**Data availability statement:** All relevant data are within the manuscript and its Supporting Information files.

**Funding:** The author(s) received no specific funding for this work.

**Competing interests:** The authors have declared that no competing interests exist.

**Abbreviations:** BCVA = best-corrected visual acuity, DM = diabetes mellitus, PVA = presenting visual acuity, SSA = sub-Saharan Africa, VI = visual impairment, VA = visual acuity.

## Results

Only 26 studies with 12508 participants, contributing data from 14 countries within Sub-Saharan Africa, met the inclusion criteria for the final analysis. The overall pooled prevalence of visual impairment was found to be 29% (95% CI: 22%−35%). Heterogeneity was observed among included studies ($p < 0.001$ for high $I^2$ values). Subgroup analysis revealed the source of heterogeneity in the studies carried out in Ethiopia ($I^2 = 99.2\%$, p =<0.001), Nigeria ($I^2 = 94.59\%$, p =<0.001), and Zambia ($I^2 = 77.34\%$, p=0.036).

## Conclusion

The findings of this study indicated that the pooled prevalence of vision impairment among diabetes mellitus patients is relatively higher. It can be concluded that diabetes mellitus patients should be managed properly to prevent visual impairment. Early detection of visual impairment through screening and regular follow-up is recommended to reduce as well as control the burden of visual impairment and its impact on the diabetic population.

## 1. Introduction

Visual impairment (VI) is a functional limitation of the eye(s) caused by a variety of medical conditions, including diabetes mellitus (DM), that cause visual field loss, decreased visual acuity, visual distortion, perceptual difficulties, or any combination of the aforementioned [1]. Some of the causes of VI include refractive error, cataracts, glaucoma, corneal opacity, age-linked macular degeneration, diabetic retinopathy, trachoma, trauma, and systemic disorders such as hyperthyroidism [2], rheumatoid arthritis, and hypertension [3].

Diabetes brings visual loss through early-onset cataracts and diabetic retinopathy, a chronic condition affecting the retinal microvasculature. Visual impairment hurts people's quality of life, mobility, vision function, emotional well-being, and social interactions, as well as being a tremendous financial burden on the worldwide economy [4,5]. Consequently, it becomes a thoughtful global health concern with significant socioeconomic and psychological effects [6]. It is believed to be more frequent in diabetic patients as opposed to non-diabetics [7].

Globally, the incidence of VI has risen dramatically. More than 2.2 billion people worldwide live with some form of visual impairment, and at least 1 billion of these cases could have been prevented or remain untreated [8]. In terms of regional variation, low- and middle-income nations are anticipated to have four times as many cases of vision impairment as high-income countries [9]. In addition, it is estimated that in Sub-Saharan Africa's (SSA) western, eastern, and central regions, the prevalence of untreated near vision impairment is estimated to be more than 80% [10]. The problem is becoming increasingly prevalent among diabetics, as diabetes becomes the major cause of VI in many affluent countries, and diabetic retinopathy, one of the complications of DM, is the most common cause of VI among those populations [11].

Although several primary studies have highlighted the burden of VI among diabetes patients in SSA, they report substantial variation in prevalence estimates and lack the robust, aggregated data necessary for effective public health planning and resource allocation across the region. To the best of our knowledge, no systematic assessment has yet been conducted to provide comprehensive data on this issue in SSA. Given the region's diverse sociodemographic landscape, limited healthcare infrastructure, and financial constraints, managing diabetes complications presents significant challenges. Understanding the burden and causes of VI in this context is critical for informing targeted interventions. Hence, this systematic review and meta-analysis aimed to determine the pooled prevalence of VI among diabetic patients in SSA, providing a scientific foundation for future research and supporting the development of evidence-based strategies for prevention and management.

## 2. Methods

### 2.1. Search strategy

This systematic review and meta-analysis was carried out according to the Preferred Reporting Items for Systematic Reviews and Meta-Analyses (PRISMA) guidelines [12] (S1 Table). The protocol was registered by the International Prospective Register of Systematic Reviews (PROSPERO) with registration number [CRD42023488655]. A literature search of the electronic databases of PubMed, Google Scholar, Web of Science, Hinay, and African Journals Online was carried out from November 12–20, 2023. The databases were searched by combining different MeSH terms and keywords, using Boolean Operators (OR, AND) (S2 Text). For example, for PubMed, the following search strategies were entered. ((((((((((((((((("vision impairment"[Title/Abstract]) OR ("Impairment, Visual"[Title/Abstract])) OR ("Impairment, Visual"[Title/Abstract])) OR ("Visual Impairments"[Title/Abstract])) OR (Micropsia[Title/Abstract])) OR (Micropsias[Title/Abstract])) OR ("Vision Disability"[Title/Abstract])) OR ("Disabilities, Vision"[Title/Abstract])) OR ("Disability, Vision"[Title/Abstract])) OR (Hemeralopia[Title/Abstract])) OR (Day Blindness[Title/Abstract])) OR ("Vision Disorder"[Title/Abstract])) OR ("Visual Disorder"[Title/Abstract])) OR (Macropsia[Title/Abstract])) OR ("Disorder, Visual"[Title/Abstract])) OR ("visual impairment"[Title/Abstract])) OR ("vision problem"[Title/Abstract]))) AND ((((("Diabetes Mellitus"[Title/Abstract]) OR ("Type1 Diabetes Mellitus"[Title/Abstract])) OR ("type 1 diabetes"[Title/Abstract])) OR ("type 2 diabetes"[Title/Abstract])) OR ("Type 2 Diabetes Mellitus"[Title/Abstract]))) AND [Country name]. The full texts of articles that appeared to meet the inclusion criteria based on the abstract were downloaded, and references were reviewed to identify additional papers that might meet the eligibility criteria.

### 2.2. Eligibility criteria

Studies were eligible for inclusion in this review if they were peer-reviewed journal articles, master's theses, or doctoral dissertations that assessed the prevalence of visual impairment (VI) and/or its causes in adult participants (aged 18 years or older, regardless of sex, diabetes type, or occupation) within community or institutional settings in SSA. To ensure the relevance and timeliness of the data, only studies published between 2010 and 2023 were considered. In addition, only articles published in English were included.

We excluded conference papers, editorials, articles without full text, and articles that had methodological problems after being reviewed by two authors using the Joanna Briggs Institute (JBI) appraisal checklist [13].

### 2.3. Data extraction and quality assessment

Two investigators (AB and WD) independently assessed the eligibility of articles and extracted data using Microsoft Excel. The data extraction sheet included the author's name, year of publication, country, study design, sample source (facility-based or population-based), number of DM patients with VI, sample size, prevalence of VI in DM, and study period. Any discrepancies between investigators were resolved by consensus. Among the articles identified, titles and abstracts were

reviewed to retrieve studies on the prevalence of VI in DM. Articles found relevant by title and abstract were screened for the full-text appraisal for eligibility. The quality of eligible studies was assessed against predefined inclusion criteria using the quality assessment tool of JBI for prevalence studies [13]. Good-quality articles were determined if the scale score was 5 and above (S3 Table).

## 2.4. Statistical analysis

Statistical analyses were done via STATA version 16 (STATA Corporation, College Station, TX, USA). The pooled prevalence of VI, accompanied by a corresponding 95% CI, was calculated using a random-effects model. Publication bias and heterogeneity were also assessed. To check publication bias, a funnel plot was made, and Egger's and Begg's tests, at a 5% significance level, were performed. The heterogeneity of studies was checked using Cochran's Q test and $I^2$ test statistics. A p-value of < 0.05 for the $I^2$ test was used to determine the presence of heterogeneity [14]. A pre-defined subgroup analysis was performed based on the country where the studies were conducted to identify the source of heterogeneity, and a P-value of <0.05 was considered significant. Furthermore, point prevalences with 95% CIs were presented in the forest plot. In this plot, each box's size indicates the study's weight, while each crossed line refers to the 95% CI. Additionally, sensitivity analysis was carried out using a leave-one-out method to look into the impact of individual studies and the stability or robustness of the pooled values to outliers.

## 2.5. Outcome of interest

The primary outcome of this study was the prevalence of VI in DM patients, which was reported in the original paper as a percentage and/or as the number of visually impaired cases (n)/total number of DM patients (N). Visual impairment was defined as any person who has poor vision or blindness [5]. The level of VI was defined according to the WHO classification, as a) normal (visual acuity (VA) 6/6–6/18)), (b) moderate VI (VA<6/18–6/60, (c) severe VI (VA worse than 6/60, but ≥3/60, and (d) very Sever VI (blindness) (VA worse than 3/60) to no perception of light [5]. The secondary outcome was the proportion of causes of VI in DM patients, calculated as a percentage of visually impaired cases due to a particular cause to the total of cases among DM patients.

## 2.6. Ethics statement

Not required as this is a systematic review and meta-analysis based on publicly available data.

## 3. Results

### 3.1. Selection and search of the studies

We initially identified a total of 590 articles in the electronic search databases of PubMed, Google Scholar, Web of Science, Hinai, and African Journals Online. The title, abstract, and full text of the articles were assessed. Of them, 288 articles were excluded due to duplicate data, 67 articles were excluded for not reporting the prevalence of VI in diabetes mellitus patients, and 197 articles were excluded for not having other required data. Finally, 26 eligible articles were selected for this meta-analysis (Fig 1).

### 3.2. Characteristics of the included studies

A total of 12508 participants were represented by 26 included studies published from 2010 to 2023. Among the studies included in the final analysis, the majority followed hospital-based cross-sectional study designs. The total number of participants per study ranged from 41 to 2320. The highest (78.05%) and the lowest (2.7%) prevalence of VI were reported in studies conducted in Ethiopia [15] and Botswana [16]. Six studies were conducted in Ethiopia [15,17–21] and four in Nigeria [22–25], (Table 1).

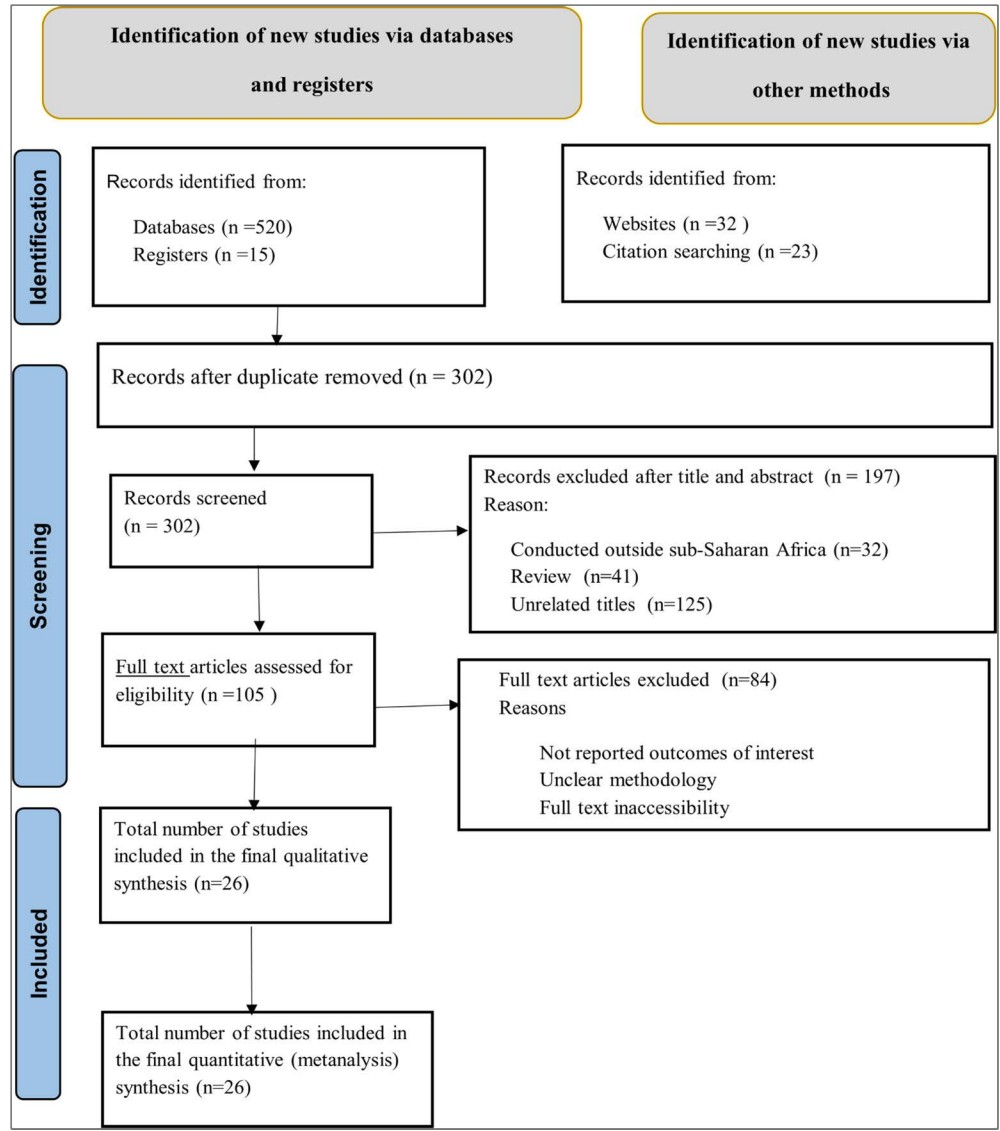

**Fig 1. Flow chart of study selection for the pooled prevalence of VI among diabetic patients in Sub-Saharan Africa.**

Regarding the definition for VI, most studies applied the WHO classification criteria [5], and 18 studies reported the severity of VI. In terms of visual acuity (VA) measurement, five studies used presenting visual acuity (PVA), 16 studies used best corrected visual acuity (BCVA), and the remaining five did not specify the type of VA measurement used (S4 Table). Based on the JBI quality assessment, all 26 primary studies scored 5 or higher, indicating moderate to high methodological quality (Table 1).

### 3.3. Prevalence of visual impairment in diabetes mellitus

A total of 3170 visually impaired diabetic patients were identified from a total of 12,508 participants in 26 included studies. The combined meta-analysis of VI in DM patients was found to be 29% (95% CI: 22%−35%). The test statistics showed

**Table 1. Summary of the 26 studies included in the meta-analysis of the prevalence of visual impairment among diabetes mellitus patients in sub-Saharan countries.**

| First author Name, Year | Country | Study population | Participant's Age, (mean ±SD) | Study design | Study Period | Sample size included | Prevalence in % with 95% CI | Quality score |
|---|---|---|---|---|---|---|---|---|
| Seid MA et al. (2022) [26] | Ethiopia | T2DM adults | *52(** [45–60]) | CS | 15 February to 15 March 2020. | 322 | 38.0 (32, 43) | 8 |
| Asemu MT, Ahunie MA (2021) [15] | Ethiopia | T1DM and T2DM | 58.9±14.6 | CS | January 2017 to January 2019. | 401 | 78.05 (74, 82) | 6 |
| Tsegaw A et al. (2021) [18] | Ethiopia | T2DM | 50.4±10.7 | CS | March 2017 to February 2018 | 719 | 26.4(23.23–29.81) | 6 |
| Alemu S., Dessie, et al. (2015) [19] | Ethiopia | T1DM | 34.5±13.3 | CS | Unspecified | 536 | 13.6 (10.7, 16.5) | 7 |
| Demilew K.Z, et al. (2022) [20] | Ethiopia | T1DM and T2DM | *45(**30–56) | CS | April 27 to May 19, 2017 | 388 | 29.4 (24.8, 33.9) | 7 |
| Alemayehu H.B, Tegegn, et al. (2022) [21] | Ethiopia | T1DM and T2DM | *49 (**40–58) | CS | May 30 to July 15, 2022 | 391 | 28. 6 (24. 2, 33.1) | 6 |
| Bastola et al. (2016) [27] | Eritrea | T1DM and T2DM | 58.8±12.9 | CS | January 2014 to October 2016 | 506 | 53.8(49.4, 58.1) | 7 |
| Glover, Burgess, et al. (2012) [28] | Malawi | T1DM and T2DM | *56.4 (**49.1–63.7 | CS | March 2007 and June 2007 | 281 | 12.1 (8.3, 15.9) | 6 |
| Burgess P.I, Allain, et al. (2014) [29] | Malawi | T1DM and T2DM | *54.1 (**43.8–61.1) | Unspecified | Unspecified | 357 | 8.1 (5.3, 11.0) | 7 |
| Awadalla (2017) [30] | Sudan | Only T2DM | ***20–70 | CS | Unspecified | 424 | 72.6 (68.4,76.9) | 6 |
| Sube LK et al.(2020) [31] | South Sudan | T1DM and T2DM | 51.1±10.67 | CS | Unspecified | 216 | 31.9(25.7, 38.2) | 8 |
| Chibuga, Bugimbi. (2012) [32] | Tanzania | T1DM and T2DM | 52.21±8.29 | CS | August to October 2012 | 163 | 23.30(17.05–30.56) | 8 |
| Seba EG, Arunga S (2015) [33] | Uganda | T1DM and T2DM | 56 | CS | September 2013 and March 2014 | 318 | 17.0(12.9, 21.1) | 8 |
| Magan.T (2019) [34] | Uganda | T1DM and T2DM | 50.4±12.5 | CS | April 16 | 41 | 14.6(3.8, 25.5) | 5 |
| Lartey SY, Aikins AK.(2018) [35] | Ghana | T1DM and T2DM | 57.5±13 | CS | Unspecified | 208 | 12.5 (8.0, 17.0) | 6 |
| Lewis A.D et al. (2018) [36] | Zambia | T1DM and T2DM | 56±11 | CS | February to June 2012 | 2153 | 17.14(15.57–18.8) | 6 |
| Patel.V. (2019) [37] | Zambia | T1DM and T2DM | *53(**17–82) | CS | December, 2018 to April, 2019 | 213 | 23.4(17.95–29.75) | 6 |
| Omari S.N (2017) [16] | Botswana | T1DM and T2DM | 56.0±13.8 | CS | January to February 2015. | 220 | 2.70(1.0–5.84) | 7 |
| Ayukotang E.N et al.(2016) [38] | Cameroon | T1DM and T2DM | 56.65±12.63 | CS | Unspecified | 84 | 22.6(13.7, 31.6) | 8 |
| Jingi, Nansseu et al. (2015) [39] | Cameron | T1DM and T2DM | 54.2±11.2 | CS | Unspecified | 407 | 29.7(25.3, 34.2) | 6 |
| Onakpoya, Adeoye, et al. (2010) [22] | Nigeria | Only T2DM | 57.5±10.8 | CS | Unspecified | 83 | 18.1(9.8, 26.4) | 7 |
| Onakpoya. O H, et al. (2015) [23] | Nigeria | Only T2DM | 61±11.8 | CS | Unspecified | 133 | 24.1(16.8, 31.3) | 7 |
| Sada B.K et al (2021) [24] | Nigeria | T1DM and T2DM | 53.5±12.3 | CS | Unspecified | 236 | 51.0(45.01–56.89) | 6 |
| Ajayi. I.A (2016) [25] | Nigeria | T1DM and T2DM | 60.42±11.72 | CS | Unspecified | 296 | 40.0(3.40, 45.1) | 7 |

*(Continued)*

**Table 1.** (Continued)

| First author Name, Year | Country | Study population | Participant's Age, (mean ±SD) | Study design | Study Period | Sample size included | Prevalence in % with 95% CI | Quality score |
|---|---|---|---|---|---|---|---|---|
| Cleland, Charles R., et al. (2016) [40] | Tanzania | T1DM and T2DM | 60.8 | CS | November 2010 to December 2014 | 3187 | 14.0(13,16) | 6 |
| Mabaso and Oduntan. (2014) [41] | South Africa | T1DM and T2DM | 61.5±10.49 | CS | May to December 2011 | 225 | 44.9(38.4, 51.4) | 6 |

CS = cross-sectional, T1DM = type 1 diabetes mellites, T2DM = type 2 diabetes mellites.

*median age, ** range.

a high heterogeneity among the studies ($I^2$ = 98.95%, p = <0.001), suggesting the application of the random effects model (Figs 2–5).

**3.3.1. Subgroup analysis.** Given the substantial heterogeneity identified among the primary studies, we conducted a subgroup analysis by country where the study was conducted and the method of visual VA measurement used to examine its potential sources. However, the source of heterogeneity was not handled. When analyzing the results by country, the highest pooled prevalence of visual impairment (VI) was observed in Sudan at 72.6% (95% CI: 68.4, 76.9), followed by Eritrea, at 53.8% (95% CI: 49.4, 58.1). Moreover, the combined prevalence of VI was 35.6% (95% CI: 17.8, 53.4) in Ethiopia, 33.3% (95% CI: 18.8, 47.8) in Nigeria, and 19.7% (95% CI: 13.6, 25.8) in Zambia. A high level of heterogeneity was observed among the studies conducted in Ethiopia ($I^2$ = 99.2%, p = <0.001), Nigeria ($I^2$ = 94.59%, p = <0.001), and Zambia ($I^2$ = 77.34%, p = 0.036) (Table 2).

Subgroup analysis based on the method of VA measurement showed that 43% of studies used presenting visual acuity (PVA), 22% used best corrected visual acuity (BCVA), and 37% did not specify their VA assessment method. There was also significant heterogeneity among the groups (Fig 6).

**3.3.2. Sensitivity analysis.** A sensitivity analysis utilizing the random-effects model was conducted to examine the impact of individual studies on the pooled prevalence of VI among DM patients in SSA. The findings indicated that no study had a significant influence on the pooled estimated effect size. When the individual studies were omitted, the pooled estimated prevalence fell within the 95% CI of the total effect size. Our pooled estimated prevalence of VI varied between 27.29% and 33.5% after the deletion of a single study (S5 Fig).

**3.3.3. Publication bias.** The symmetrical distribution of the funnel plot demonstrated that there was no publication bias among the included studies upon visual inspection of the funnel plot (Fig 7).

## 3.4. Causes of visual impairment

Only 9 studies reported the causes of VI in diabetic patients [17,20–23,25,32,39], and the combined prevalence was estimated from those studies. Accordingly, the most common cause of VI was found to be diabetic retinopathy, representing 35% (95% CI: 16, 35) followed by cataracts at 34% (95% CI: 21.0, 48.0) and diabetic maculopathy representing 26.0% (95% CI: 11.0, 41) (S6 Fig).

## 4. Discussion

This systematic review and meta-analysis investigated the prevalence of visual impairment among individuals with diabetes mellitus in sub-Saharan Africa, revealing a pooled prevalence of 29% (95% CI: 22%–35%). Further disaggregation of the data was done by severity of visual impairment, revealing pooled prevalence estimates of 15.7% for moderate VI (95% CI: 11.7–20.1%), 5.0% for severe VI (95% CI: 3.3–7.0%), and 7.0% for blindness (95% CI: 3.0–10.0%). These findings suggest that a significant proportion of individuals with diabetes in SSA experience serious visual deficits, which can impair quality of life, limit productivity, and increase dependency [42,43].

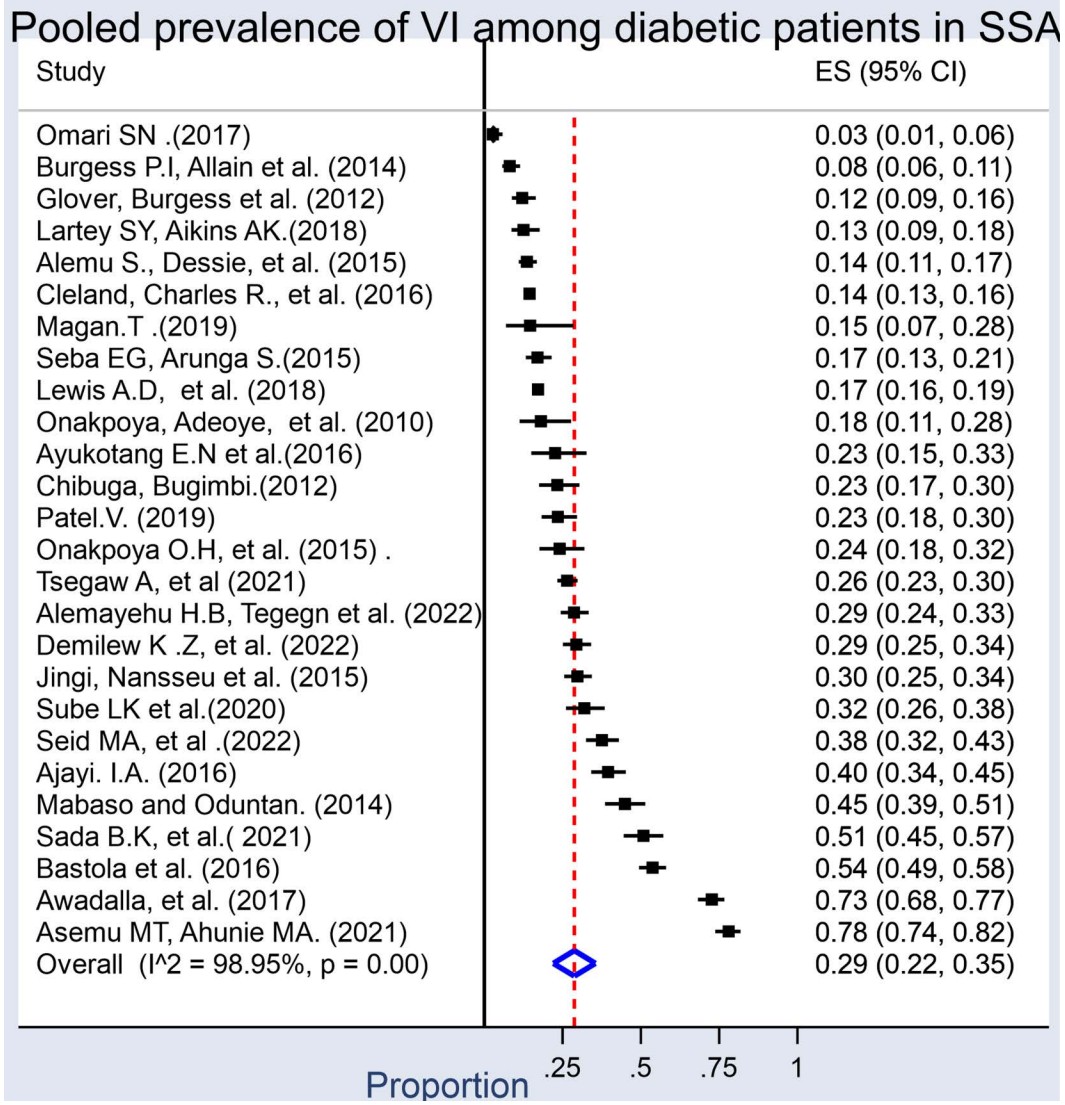

**Fig 2. Forest plot depicting the pooled prevalence of visual impairment among diabetes mellitus patients in sub-Saharan Africa.** Pooled analysis of the level of VI has also been conducted for 18 studies to determine the severity of VI. As a result, the overall prevalence of moderate VI was 15.7% (95% CI: 11.7%–20.10%, I² = 98.85%, p-value<0.001).

The finding in this study is considerably higher than those reported in other regions. For instance, a global meta-analysis of population-based studies between 1990 and 2010 reported the prevalence of moderate to severe VI in diabetic patients as 1.9% globally, 2.8% in Central Asia, and 2.5% in Central Europe [44]. This discrepancy could, in part, be attributed to differences in the timeframe of the studies; the earlier global analysis reflects an older period, which may have had different diagnostic capabilities, screening uptake, or a less pronounced diabetes epidemic than the 2010–2023 period covered by our study. Moreover, variability in the operational definitions of VI (e.g., acuity thresholds or inclusion criteria) may have impacted prevalence estimates significantly.

Similarly, a global systematic review and meta-analysis on the prevalence of moderate to severe vision impairment due to diabetic retinopathy among older adults in 2015 reported a worldwide average of just 1.30% [45]. In regional

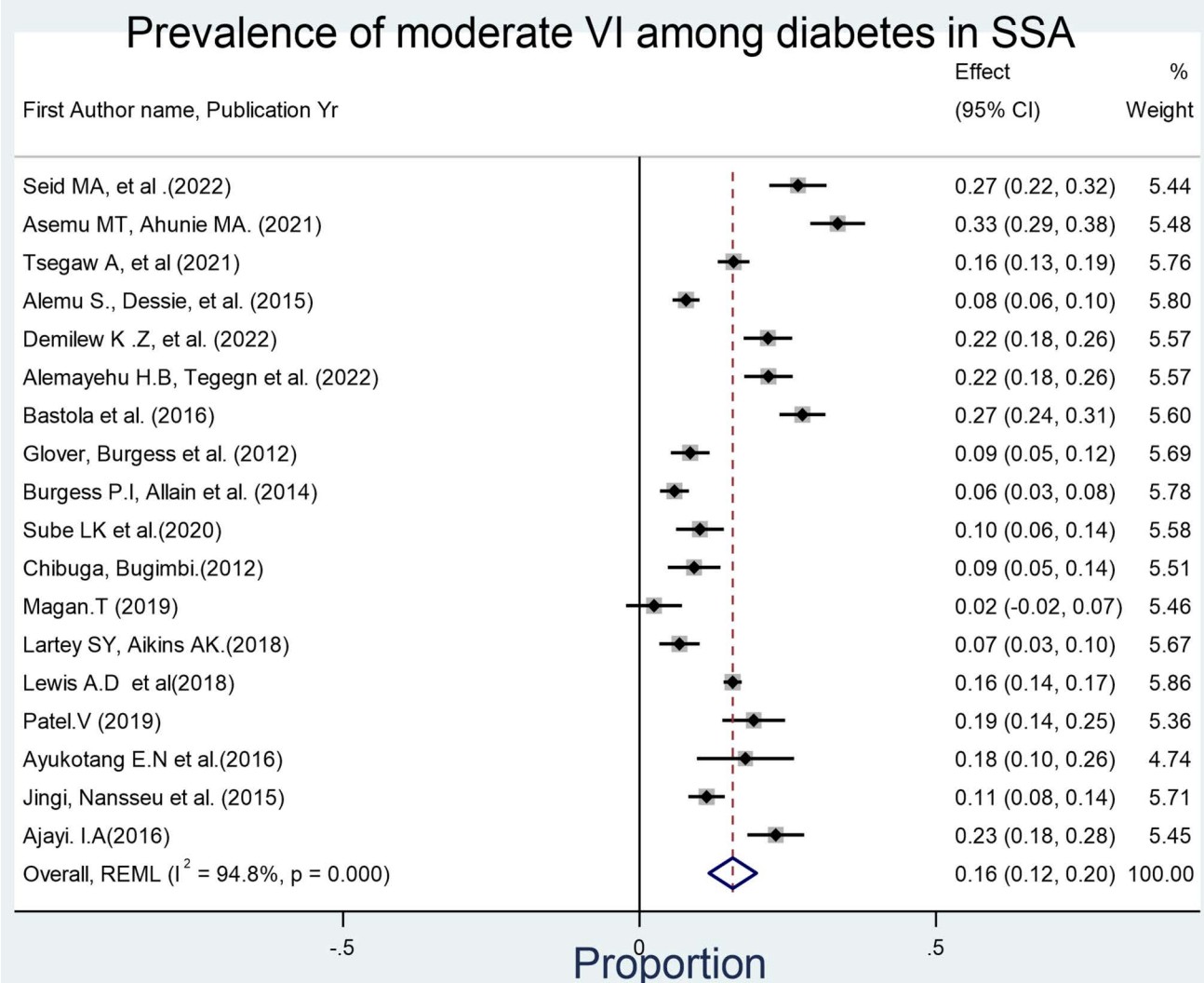

**Fig 3. Forest plot showing the overall prevalence of Moderate visual impairment (VI) among diabetes mellitus patients in sub-Saharan Africa.**
Similarly, the pooled prevalence of severe VI and very severe VI (blindness) was found to be 5.1% (95% CI: 3.3%−7.00%, I²=92.5%, p-value<0.001).

breakdowns, Southern Latin America showed 3.93%, Central Asia 4.06%, and Eastern Europe 5.06%. Even in high-income regions, such as Asia Pacific (4.07%) and Australasia (4.72%) [46], the prevalence remains significantly lower than the current study estimate. While the global review focused specifically on diabetic retinopathy-related VI, this meta-analysis considered all-cause of VI among DM patients. This broader scope, particularly in a region with limited access to comprehensive eye care, highlighes a disproportionately high burden of VI in SSA and points to avoidable causes that could significantly reduce blindness and disability [47].

The higher burden of VI among DM patients in SSA is further supported by a previous estimate, which reported the percentage of total moderate to severe VI for all ages due to diabetic retinopathy as only 1.9% worldwide, reaffirming the low global baseline when compared to our 29% pooled prevalence for SSA [48]. These disparities emphasize that while diabetic complications may be more effectively managed or detected in other regions, SSA may face a confluence of risk factors, including limited access to diabetic eye care, late diagnosis, and poor glycemic control, among others [49–51].

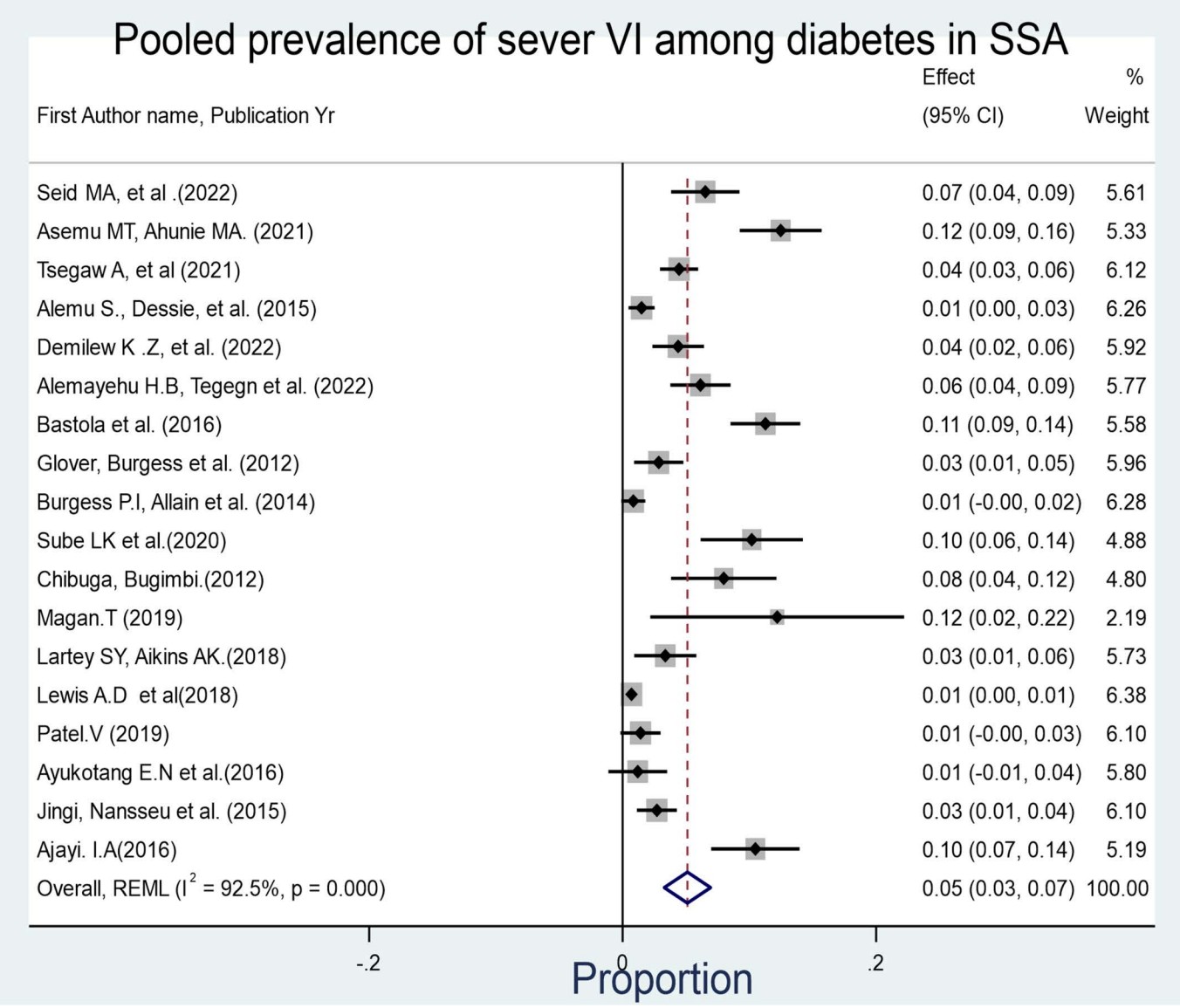

**Fig 4. Forest plot depicting the allover prevalence of severe visual impairment among diabetes mellitus patients in Sub-Saharan Africa and 7.0% (95% CI: 3.0%−10.02%, I² =95.9%, p-value<0.001) respectively.**

Multiple systemic and health-related factors could further contribute to this disparity. First, access to diabetic eye care services remains limited in many SSA countries, and there is often a shortage of trained ophthalmologists and retinal specialists, particularly in rural areas [51]. Second, the delayed diagnosis of both diabetes and its ocular complications is common due to weak screening systems and poor health-seeking behavior, resulting in more advanced disease stages at presentation [52]. Third, poor glycemic control, often due to cost barriers to medication, lack of dietary counseling, and inadequate follow-up care, accelerates the onset and progression of diabetic eye disease [51,53]. These factors are exacerbated by low awareness of DR and its risks among patients and primary healthcare providers. Finally, the absence of systemic eye screening programs in many SSA countries contrasts sharply with protocols in high-income settings, where

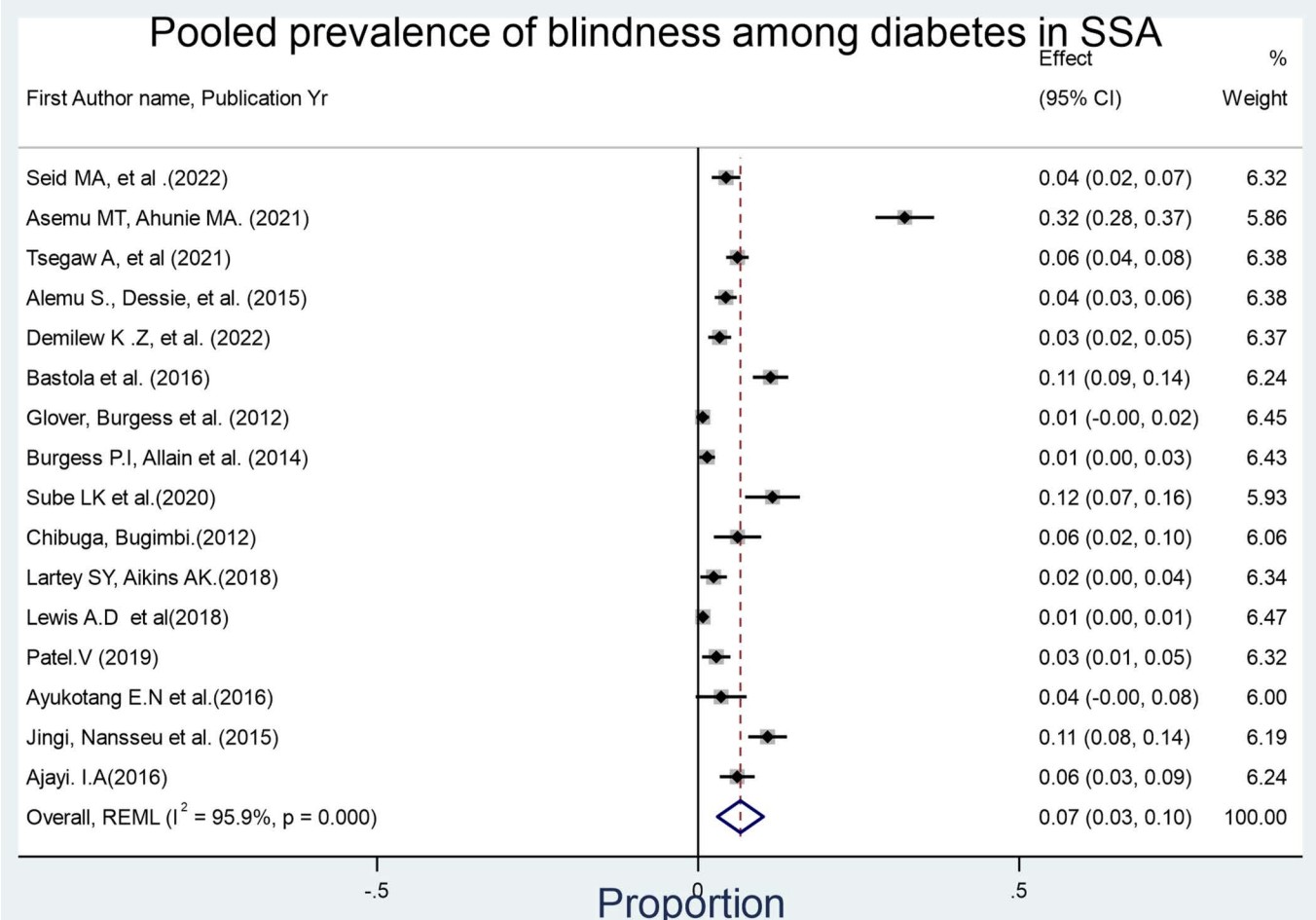

**Fig 5. Forest plot depicting the pooled prevalence of blindness among diabetes mellitus patients in Sub-Saharan Africa.**

regular eye exams for diabetic patients are standard practice [51,54]. This gap delays detection and intervention, leading to a higher incidence of irreversible vision loss.

Subgroup analysis by country revealed substantial variability in prevalence estimates of VI, ranging from as low as 2.7% in Botswana to 53.8% in Eritrea and 72.6% in Sudan. This heterogeneity may stem from various factors, including differences in VI definitions, study settings [55], differences in population prevalence of DM and ocular complications [56], as well as socioeconomic and demographic disparities [57]. The prevalence of ocular complications such as diabetic retinopathy (DR) varies across SSA. For instance, a study in Eritrea [27] found a DR prevalence of 84% among diabetic patients, indicating its significant burden. Higher DR rates are linked to increased risk of visual impairment and contribute to variability in VI estimates across countries [58]. Additionally, disparities in healthcare infrastructure, access to screening services, and population-level diabetes control measures likely contribute to the observed variation. Botswana, for example, has made considerable investments in its healthcare system and has relatively better access to diabetic care and eye health services. These developments likely contribute to earlier detection and management of diabetic complications, including diabetic retinopathy, thereby lowering the prevalence of visual impairment [59,60]. The differences in the underlying prevalence of diabetes and associated comorbidities across these countries further complicate direct comparisons and reinforce the need for localized data to inform tailored responses.

**Table 2. Sub-group analysis showing the prevalence of visual impairment among patients with diabetes mellitus in sub-Saharan Africa, 2023.**

| Sub-group | | Number of included studies | Prevalence (95% CI) | Heterogeneity statistics | |
|---|---|---|---|---|---|
| | | | | I² | p-value |
| By country | Sudan | 1 | 72.6%(68.4, 76.9) | 0.00% | |
| | Botswana | 1 | 2.7%(0.6, 4.9) | 0.00% | |
| | Cameron | 2 | 27.3%(20.7, 33.9) | 48.64% | 0.163 |
| | Erterea | 1 | 53.8%(49.4, 58.1) | 0.00% | |
| | Ethiopia | 6 | 35.6%(17.8, 53.4) | 99.2% | <0.001 |
| | Ghana | 1 | 12.5%(8.0, 17.0) | 0.00% | |
| | Malawi | 2 | 9.9%(6.0, 13.8) | 62.84% | 0.101 |
| | Nigeria | 4 | 33.3%(18.8, 47.8) | 94.59% | <0.001 |
| | South Africa | 1 | 44.9%(38.4, 51.4) | 0.00% | |
| | South Sudan | 1 | 31.9%(25.7, 38.2) | 0.00% | |
| | Tanzania | 2 | 18.3 (9. 7, 26.9) | 85.49% | |
| | Uganda | 2 | 16.7%(12.8, 20.5) | 0.02% | 0.691 |
| | Zambia | 2 | 19.7%(13.6, 25.8) | 77.34% | 0.036 |

Notably, subgroup analysis based on the method of VA measurement revealed that 43% of studies used presenting visual acuity (PVA), 22% used best corrected visual acuity (BCVA), and 37% did not specify their VA assessment method. There was significant heterogeneity among these groups, suggesting that differences in measurement approach could contribute to variability in reported prevalence [61,62]. Presenting visual acuity may overestimate VI due to uncorrected refractive errors, while BCVA provides a more conservative estimate by accounting for correctable causes [62]. The inconsistency in VA assessment methods indicates the need for standardized protocols in epidemiological studies to enable more accurate comparisons across settings. This variability in VA measurement methods can significantly impact prevalence estimates and affect direct comparisons between studies [63].

In the current study, the common causes of visual impairment (VI) were diabetic retinopathy (35%, 95% CI: 16.0–35.0), followed by cataracts (34%, 95% CI: 21.0–48.0), and diabetic maculopathy (26%, 95% CI: 11.0–41.0). These results are consistent with previous studies indicating that DR is a leading cause of vision loss in patients with diabetes [64]. Diabetic retinopathy is characterized by progressive retinal microvascular damage due to chronic hyperglycemia, resulting in increased vascular permeability, retinal ischemia, and ultimately, neovascularization and vision loss [65]. Cataracts, another common complication, develop earlier and more frequently in diabetic individuals due to metabolic disturbances in the lens [66]. Diabetic maculopathy also contributes significantly to VI in SSA, highlighting the multifaceted impact of diabetes on ocular health [67]. Other causes, including glaucoma, were less commonly reported but remain clinically relevant [68].

Early diagnosis and management of these ocular complications are crucial for preventing vision loss. Recent advances in imaging modalities, such as fundus photography and optical coherence tomography, have significantly improved the ability to detect early retinal changes and monitor disease progression [69,70]. However, such technologies remain underutilized in many SSA settings due to cost and limited access. Strengthening routine screening programs and integrating ophthalmic services into diabetes care pathways can help address this gap and reduce preventable vision loss.

## 5. Strengths and limitations of the study

As a strength, this study employed comprehensive search strategies across multiple databases to capture all available literature on the prevalence of visual impairment among diabetes mellitus patients in sub-Saharan Africa. However, it has several limitations. First, it excluded studies published in languages other than English, which may have missed relevant

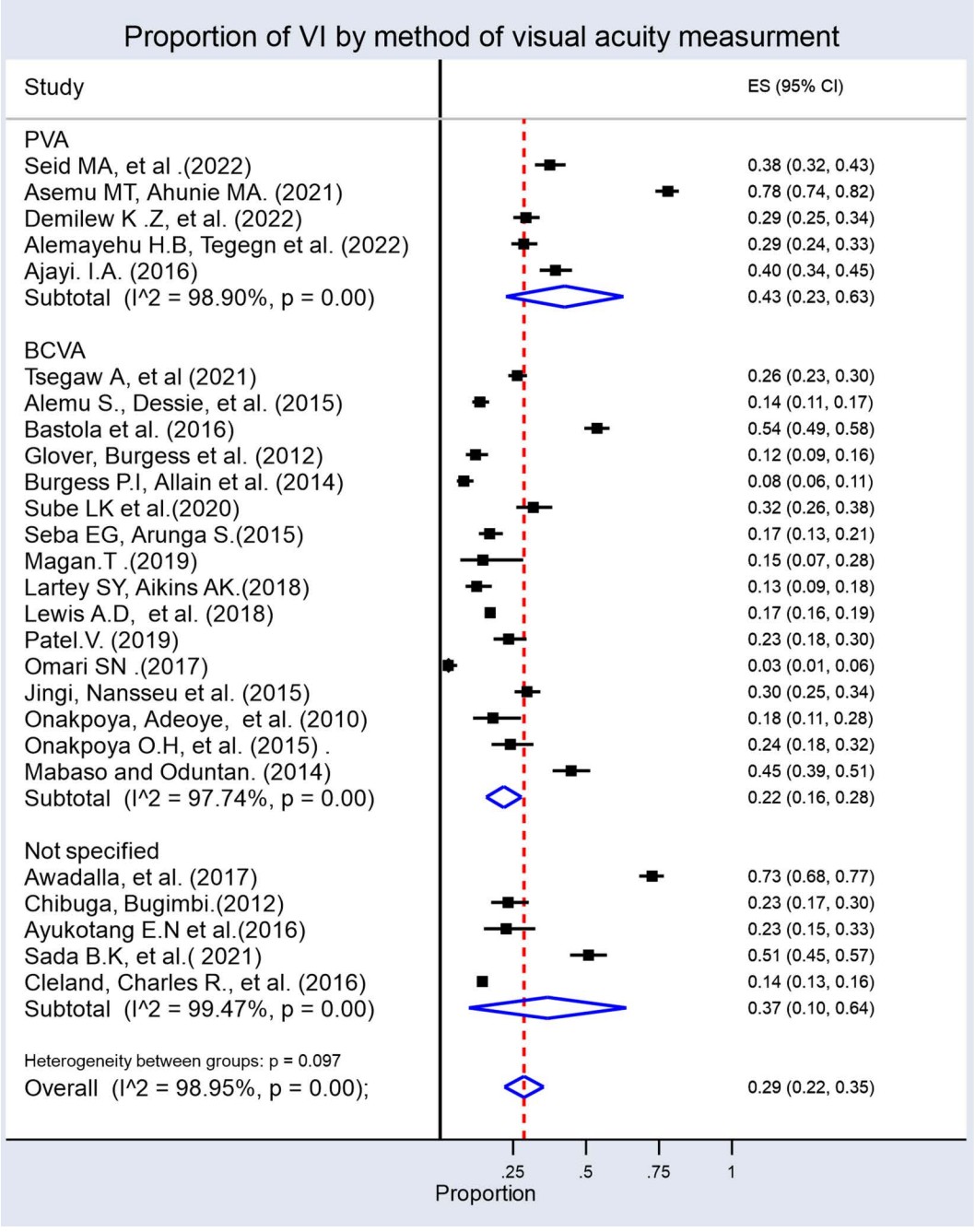

**Fig 6. Sub-group analysis by the method of visual acuity measurement showing the prevalence of VI among patients with diabetes mellitus in sub-Saharan Africa.**

data. Second, only fourteen countries within the region were represented, suggesting that many others may be under-represented. Third, the study did not identify predictors of visual impairment among diabetic patients due to the limited number of relevant studies. Fourth, variability in visual acuity measurement, including unclear specification of whether best-corrected or uncorrected acuity was used, which may introduce variability in the interpretation of visual impairment

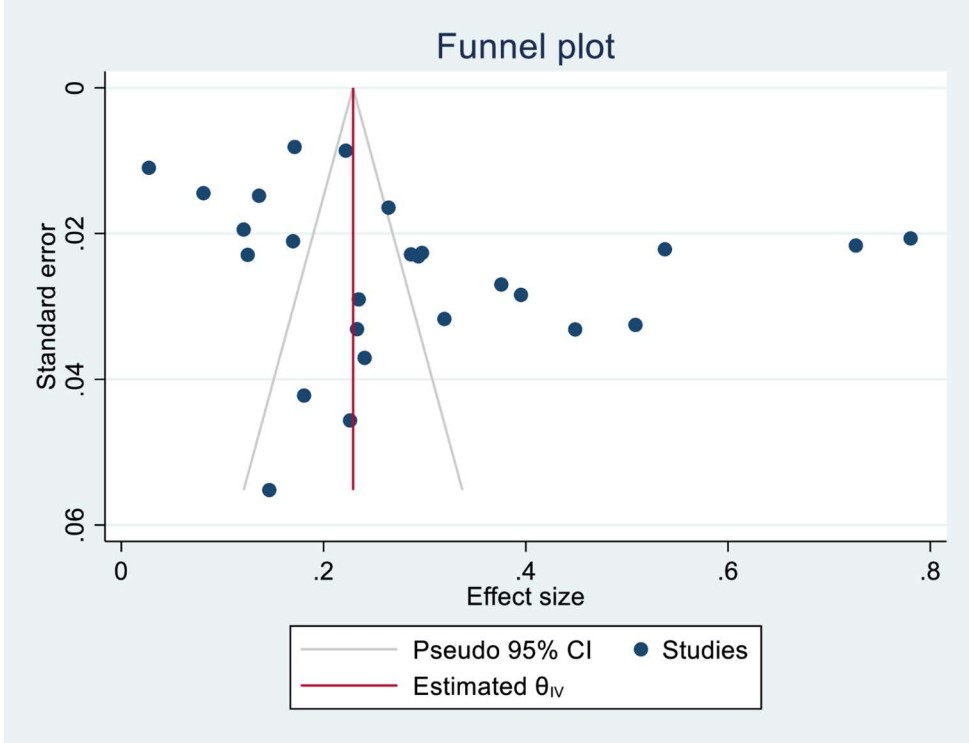

**Fig 7. Funnel plot of the risk of publication bias for the prevalence of visual impairment among diabetic patients in Sub-Saharan Africa.** Egger's test was also conducted, and the result showed no evidence of publication bias, with a p-value of 0.6261.

across studies, may affect the consistency of findings. Lastly, since the included studies were conducted in healthcare facilities, the results may overestimate prevalence and may not accurately reflect the true population-level burden.

## 6. Conclusion

In conclusion, this study showed that the magnitude of VI was somewhat high among DM patients in SSA. It varies in prevalence throughout the countries in the region, with Sudan having the highest prevalence. Diabetic retinopathy and cataracts were the most common causes. These findings reinforce the need for improved diabetes care, regular eye screening, and accessible ophthalmologic services across the region. Furthermore, situation-based interventions and country context-specific preventive strategies should be planned to lessen the burden of VI. Finally, it is advised to closely monitor the patient and use the appropriate preventative measures.

## Supporting information

**S1 Table.  PRISMA_2020_checklist.**
(DOCX)

**S2 Text.  Online database searching strategies/MeSH terms and entry terms.**
(DOCX)

**S3 Table.  JBI critical appraisal checklist for included prevalence studies.**
(DOCX)

**S4 Table. Table summarizing visual acuity measurement.**
(DOCX)

**S5 Fig. Sensitivity analysis for the studies included in the prevalence of visual impairment among diabetes.**
(DOCX)

**S6 Fig. Forest plots showing causes of visual impairment among patients with diabetes mellitus, in sub-Saharan Africa.**
(DOCX)

**S7 Table. Table summarizing all data extracted from the primary research source.**
(XLS)

**S8 Data. Data set.**
(DTA)

**S9 Table. List of all studies identified in the literature search.**
(DOCX)

## Author contributions

**Conceptualization:** Altaseb Beyene Kassaw, Addis Alem Hadigu, Wondwossen Debebe, Gosa Mankelkl, Alemu Gedefie.

**Data curation:** Natnael Kebede, Alemu Gedefie.

**Formal analysis:** Natnael Kebede, Alemu Gedefie.

**Methodology:** Altaseb Beyene Kassaw, Wondwossen Debebe, Natnael Kebede, Alemu Gedefie.

**Software:** Natnael Kebede.

**Supervision:** Altaseb Beyene Kassaw, Addis Alem Hadigu, Melese Shenkut Abebe, Amare Abera Tareke.

**Validation:** Gosa Mankelkl.

**Writing – original draft:** Altaseb Beyene Kassaw, Wondwossen Debebe.

**Writing – review & editing:** Altaseb Beyene Kassaw, Melese Shenkut Abebe, Amare Abera Tareke, Wondwossen Debebe, Gosa Mankelkl, Natnael Kebede, Alemu Gedefie.

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
