## [Decision Letter · Decision Letter 0]

Dear Dr. Kassaw,

Thank you for submitting your manuscript to PLOS ONE. After careful consideration, we feel that it has merit but does not fully meet PLOS ONE’s publication criteria as it currently stands. Therefore, we invite you to submit a revised version of the manuscript that addresses the points raised during the review process.

We look forward to receiving your revised manuscript.

Kind regards,

Tebelay Dilnessa, MSc

Academic Editor

PLOS ONE

2. In the online submission form you indicate that your data is not available for proprietary reasons and have provided a contact point for accessing this data. Please note that your current contact point is a co-author on this manuscript. According to our Data Policy, the contact point must not be an author on the manuscript and must be an institutional contact, ideally not an individual. Please revise your data statement to a non-author institutional point of contact, such as a data access or ethics committee, and send this to us via return email. Please also include contact information for the third party organization, and please include the full citation of where the data can be found.

3. As required by our policy on Data Availability, please ensure your manuscript or supplementary information includes the following:

Additional Editor Comments:

Use of English language is poor in certain sections and would require a detailed revision.All figures should be attached as separate files in TIF version.All supplementary files title/description should be written below reference listsAll authors’ email except the corresponding author should be removed from the main manuscript.The author should follow uniform font size, font type, paragraphing, etc.Follow the standard binomial nomenclature, italize journal name and the word ‘et al’

Reviewers' comments:

Reviewer's Responses to Questions

**Comments to the Author**

1. Is the manuscript technically sound, and do the data support the conclusions?

Reviewer #1: Yes

Reviewer #2: Partly

2. Has the statistical analysis been performed appropriately and rigorously?

Reviewer #1: Yes

Reviewer #2: Yes

3. Have the authors made all data underlying the findings in their manuscript fully available?

Reviewer #1: Yes

Reviewer #2: Yes

4. Is the manuscript presented in an intelligible fashion and written in standard English?

Reviewer #1: No

Reviewer #2: No

Reviewer #1: Dear Editor,

I appreciate the opportunity to review the manuscript titled " Prevalence of Vision Impairment among Patients with Diabetes Mellitus in Sub-Saharan Africa: a Systematic Review and Meta-Analysis from 2010-2023" submitted to Plos One. Below, I have provided a detailed critique of the manuscript and recommendations for improvement.

General Assessment

The manuscript addresses an important and relevant topic in diabetes management, providing potential contributions to consider optimal control of sugar control to reduce visual impairment as a main complication However, the significant spelling and grammatical errors throughout the manuscript require attention before the manuscript can be considered for publication.

Abstract

Results

The overall pooled prevalence of visual impairment was found to be 29.1% (95% CI: 22%- 36.2%).

The overall prevalence in the forest plot is 29%. Why did you say 29.1% in the abstract?? Please countercheck.

Introduction

Our goal in performing this systematic review and meta-analysis is there for to offer a comprehensive picture of the prevalence of visual impairment and its causes in patients with DM, which will help guide the development of evidence-based management and preventive plans.

Line 88: Please correct the spelling error: “there for” to therefore

Methods

Can you clarify, why you only consider studies from 2010-2023?

Eligibility Criteria

Why did you include master's thesis, and dissertations? I think It is less frequently considered in systematic reviews and meta-analyses. Did carefully evaluate their quality and relevance due to potential issues like lack of peer review?

Language: Did you exclude several articles Articles published in other languages?

How did you handle studies reporting visual impairment that was diagnosed concurrently with diabetes? Did you encounter this kind of study during the screening?

Results

Characteristics of the Included Studies

A total of 11240 participants were represented by 26 included studies published from 2010 to 2022. I think 2022 should be 2023. Please counter check

Six studies were conducted in Ethiopia (16, 18-22) and foure 182 in Nigeria(23-26), (Table 1). ‘’Foure’’ should be changed to’’ four’’

Table 1. Summary of 25 studies included in the meta-analysis of the prevalence of Vi among diabetes mellitus patients in Sub-Saharan Countries, 2022.

Vi should be changed to VI. If possible, please kindly avoid the use of abbreviations in Figure and Table titles. Remove 2022 in the title as well.

Aprivations: CS= cross-sectional, T1DM = Type 1 diabetes mellites, T2DM= Type 2 diabetes mellites: Please correct the spelling errors

3.3. Prevalence of Visual Impairment in Diabetes Mellitus

A total of 3,224 visualy impaired diabetic patie were identified from a total of 11, 240 194 participants in 26 included studies The combined meta-analysis of VI in DM patients was found 195 to be 29.1% (95% CI: 22%- 36.2%). Please correct the spelling and punctuation error in this statement.

As a result, the all over prevalence of moderate VI was 15.7% ( 95% CL: 11.7%-20.10%, I 2 199 =98.85%, p-value. Please correct the spelling error. I think the most common terminology is confidence interval, not confidence level

Please have the spacing in the title of the forest plot figure.

Fig.4. Forest plot depicting the allover prevalence of sever visual impairment among diabetes 212 mellitus patients in Sub-Saharan Africa, 2023

Pls check the spelling error of severe and change allover to overall.

When the individual studies were omitted, the pooled estimated prevalence falls within the 95 % CI of the total effect size . Our pooled estimated prevalence of VI varied between 27.29% and 33.5%) after deletion of a single study (Supplementary file 5).

Please correct as follows:

When the individual studies were omitted, the pooled estimated prevalence fell within the 95 % CI of the total effect size. Our pooled estimated prevalence of VI varied between 27.29% and 33.5%) after the deletion of a single study (Supplementary file 5).

It will be good if you manage to anayze the disparities in visual impairment between type I and type 2 Diabetes for studies reporting the prevalence of visual impairment in both types of diabetes

3.4. Causes of Visual Impairment.

Accordingly, the most caus of VI was 245 found to be diabetic retinopathy, representing 35%( 95% CL: 16, 35) followed by cataract 246 34%(95% CL: 21.0, 48.0) and diabetic maculopathy representing 26.0% ( 95% CL: 11.0, 41) (supplementary file 5). Please correct the spelling error ‘Cause ‘

Discussion

Can you look for similar reports in Western countries for comparison?

• What is the plausible explanation for the highest prevalence of visual impairment in Sudan?

References

Some of the reference lists did not have the complete list of information required for the referencing style you used. Please review and revise it accordingly.

Reviewer #2: Overview:

The authors present the prevalence of visual impairment among diabetic patients in Sub-Saharan Africa based on a a systematic review and meta-analysis. However, I have major concerns regarding the study design and measurement of vision in the reviewed studies which may heavily impact the conclusions of the current study.

Major comments:

You report that “the majority (of reviewed studies) were followed hospital-based cross-sectional study designs”. This could be a major limitation to the study, considering the difficulties to generalize hospital-based results to population level. This could also explain the particularly high prevalence of visual impairment in the diabetic patient samples.

There is no description of how visual acuity has been measured in each study, whether it was best-corrected visual acuity, and whether there were significant differences in the measurements between studies. This could be a second major limitation to the study.

There are also multiple grammatical errors and typos throughout the text. I highly recommend re-checking the manuscript thoroughly for these and consult an English language expert if needed.

Minor comments:

Consider removing the WHO classification of visual impairment from the Introduction as it is more properly included in the Materials and Methods.

Uniform the presentation of the data in Table 1 for better clarity.

Recent population-based studies have reported a significant decrease in visual impairment among diabetic patients (see references below). Considering the differing results in the current study, you should add more discussion about this difference.

Laatikainen et al. 2016 Improving visual prognosis of the diabetic patients during the past 30 years based on the data of the Finnish Register of Visual Impairment

Purola et al. 2022 Changes in Visual Impairment due to Diabetic Retinopathy During 1980–2019 Based on Nationwide Register Data

**Do you want your identity to be public for this peer review?** For information about this choice, including consent withdrawal, please see our Privacy Policy

Reviewer #1: **Yes: ** Amsalu Degu

Reviewer #2: No

---

## [Author Response · Author response to Decision Letter 1]

3 Mar 2025

Dear Editors and Reviewers,

We sincerely appreciate your time and effort in reviewing our manuscript. We have carefully considered each comment and made the necessary revisions accordingly. Below, we provide our detailed responses to each point raised.

Responses to Editor’s Comments

1. Please ensure that your manuscript meets PLOS ONE's style requirements, including those for file naming. The PLOS ONE style templates can be found at…

Response: Many thanks for your kind reminder. We have carefully reviewed the PLOS ONE style requirements and reformatted our manuscript to ensure full compliance.

2. [Data Availability Statement]

Response: We apologize for the error in our initial submission. We have now updated the data availability statement to reflect that the data is available. The data can be accessed upon request, and we are prepared to provide it whenever required. Please let us know if any further modifications are needed to meet the journal’s requirements.

3. -a) A numbered table of all studies identified in the literature search, including excluded studies with reasons for exclusion. A table of all extracted data from primary research sources, ensuring that all necessary details are available for replication].

Response: We have provided a numbered table of all studies included in the final analysis (Table 1). The PRISMA flow diagram (Figure 1) illustrates the total number of studies identified through the literature search, the study selection process, including the excluded studies and the reasons for their exclusion. However, if this does not fully meet the journal's requirements, we are happy to provide a separate numbered table of all studies identified in the literature search, including those that were excluded from the analysis, along with the specific reasons for their exclusion. Please let us know if you would like us to proceed with this additional table.

b)- A table of all data extracted from the primary research sources for the systematic review and/or meta-analysis. The table must include the following information for each study:

Response: We have provided a comprehensive table (S6) summarizing all extracted data from the primary research sources in our systematic review and meta-analysis, including the names of data extractors, extraction dates, study eligibility confirmation, and all relevant data required to replicate our analyses.

c) A table detailing the risk of bias and quality assessments for each included study.

Response: We have included a table that shows the risk of bias and quality assessments for each study included in our analysis (the JBI quality assessment tool /risk of bias assessment tool for prevalence studies (S3)).

Response: We appreciate your comment and would like to clarify that as this was a systematic review and meta-analysis, we initially excluded primary studies that did not report the outcome variable, hence there was no missing data that could affect our pooled analysis. However, if we have misunderstood your query, we would be happy to provide further clarification or additional information as required.

4. [Additional Editor Comments]:

• Use of English language is poor in certain sections and would require a detailed revision.

• All figures should be attached as separate files in TIF version.

• All supplementary files title/description should be written below reference lists

• All authors’ email except the corresponding author should be removed from the main manuscript.

• The author should follow uniform font size, font type, paragraphing, etc.

• Follow the standard binomial nomenclature, italize journal name and the word ‘et al’

Response: we appreciate your kind reminder, too. To address your comments:

1. We have carefully revised the manuscript to improve the English language in the identified sections.

2. All figures have been reattached as separate TIF files as per your request.

3. The titles/descriptions of all supplementary files have been moved to below reference list.

4. We have removed all authors’ emails except for the corresponding author from the main manuscript.

5. We have ensured consistency in font size, font type, and paragraph formatting throughout the manuscript.

6. We have, italicized the word ‘et al.'

Response to Reviewers’ Comments

Reviewer 1

[General Assessment]: The manuscript addresses an important and relevant topic in diabetes management, providing potential contributions to consider optimal control of sugar control to reduce visual impairment as a main complication. However, the significant spelling and grammatical errors throughout the manuscript require attention before the manuscript can be considered for publication

Response: Thank you for your thoughtful review of our manuscript. We appreciate your positive feedback on the relevance of the topic and acknowledge your concern about the spelling and grammatical errors throughout the manuscript. We apologize for these issues and have thoroughly revised the manuscript to correct all errors and improve clarity. We believe these revisions have enhanced the quality of the manuscript, and we are grateful for your valuable suggestions.

[Abstract]:

Results

The overall pooled prevalence of visual impairment was found to be 29.1% (95% CI: 22%- 36.2%).

The overall prevalence in the forest plot is 29%. Why did you say 29.1% in the abstract?? Please countercheck.

Introduction

Our goal in performing this systematic review and meta-analysis is there for to offer a comprehensive picture of the prevalence of visual impairment and its causes in patients with DM, which will help guide the development of evidence-based management and preventive plans.

Line 88: Please correct the spelling error: “there for” to therefore.

Response: Thank you for your kind reminder. We apologize for the discrepancy between the forest plot and the result in the abstract regarding the pooled prevalence. Upon rechecking, we corrected to 29%.

Introduction [Spelling Error]: Line 88: Please correct the spelling error: “there for” to therefore.

Response: Thank you for pointing out this. We have corrected "there for" to “therefore” in the revised manuscript.

[Methods]

Can you clarify, why you only consider studies from 2010-2023?

Eligibility Criteria

Why did you include master's thesis, and dissertations? I think It is less frequently considered in systematic reviews and meta-analyses. Did carefully evaluate their quality and relevance due to potential issues like lack of peer review?

Language: Did you exclude several articles Articles published in other languages?

How did you handle studies reporting visual impairment that was diagnosed concurrently with diabetes? Did you encounter this kind of study during the screening?

Response: Thank you for your thoughtful and insightful questions. We appreciate your careful review of our manuscript.

- Time Frame (2010-2023): We limited the inclusion of studies to those published in this year's interval to ensure the relevance and currency of the data, as we aimed to provide an up-to-date overview of the prevalence of visual impairment in DM patients. This time frame was chosen to reflect recent burdens and to ensure the inclusion of studies that are more relevant to current clinical practices and research. Additionally, we observed that there were fewer studies published before 2010, and the majority of relevant research falls within the 2010-2023 period.

- Eligibility Criteria

o We recognize that these sources may not have undergone peer review; however, we carefully assessed their quality and relevance by evaluating the methodology, data quality, and outcomes reported. We included only those that met our eligibility criteria for methodological rigor.

o Language- Due to resource limitations for translation, we initially applied a filter in each database to include only articles published in English. As a result, articles in other languages were excluded from the review.

o During the screening process, we did not encounter any studies that clearly reported visual impairment diagnosed concurrently with diabetes. As a result, we did not need to address this scenario in our review.

[Result]

Characteristics of the Included Studies

A total of 11240 participants were represented by 26 included studies published from 2010 to 2022. I think 2022 should be 2023. Please counter check

Six studies were conducted in Ethiopia (16, 18-22) and foure 182 in Nigeria(23-26), (Table 1). ‘’Foure’’ should be changed to’’ four’’

Table 1. Summary of 25 studies included in the meta-analysis of the prevalence of Vi among diabetes mellitus patients in Sub-Saharan Countries, 2022.

Vi should be changed to VI. If possible, please kindly avoid the use of abbreviations in Figure and Table titles. Remove 2022 in the title as well.

Aprivations: CS= cross-sectional, T1DM = Type 1 diabetes mellites, T2DM= Type 2 diabetes mellites: Please correct the spelling errors

3.3. Prevalence of Visual Impairment in Diabetes Mellitus

A total of 3,224 visualy impaired diabetic patie were identified from a total of 11, 240 194 participants in 26 included studies The combined meta-analysis of VI in DM patients was found 195 to be 29.1% (95% CI: 22%- 36.2%). Please correct the spelling and punctuation error in this statement.

As a result, the all over prevalence of moderate VI was 15.7% ( 95% CL: 11.7%-20.10%, I 2 199 =98.85%, p-value. Please correct the spelling error. I think the most common terminology is confidence interval, not confidence level

Please have the spacing in the title of the forest plot figure.

Fig.4. Forest plot depicting the allover prevalence of sever visual impairment among diabetes 212 mellitus patients in Sub-Saharan Africa, 2023

Pls check the spelling error of severe and change allover to overall.

When the individual studies were omitted, the pooled estimated prevalence falls within the 95 % CI of the total effect size . Our pooled estimated prevalence of VI varied between 27.29% and 33.5%) after deletion of a single study (Supplementary file 5).

Please correct as follows:

When the individual studies were omitted, the pooled estimated prevalence fell within the 95 % CI of the total effect size. Our pooled estimated prevalence of VI varied between 27.29% and 33.5%) after the deletion of a single study (Supplementary file 5).

It will be good if you manage to anayze the disparities in visual impairment between type I and type 2 Diabetes for studies reporting the prevalence of visual impairment in both types of diabetes

3.4. Causes of Visual Impairment.

Accordingly, the most caus of VI was 245 found to be diabetic retinopathy, representing 35%( 95% CL: 16, 35) followed by cataract 246 34%(95% CL: 21.0, 48.0) and diabetic maculopathy representing 26.0% ( 95% CL: 11.0, 41) (supplementary file 5). Please correct the spelling error ‘Cause ‘.

Response: Thank you for your detailed and helpful feedback. We have carefully reviewed your comments and made the necessary corrections, as follows:

1. You are correct that the last study in our review was published in 2023, and we corrected it accordingly.

2. We have corrected the spelling of “foure” to “four”

3. Table 1 Title: We have changed “Vi” to visual impairment and removed the abbreviation, as suggested. We also took care to avoid abbreviations in other figure and table titles.

4. We have corrected the spelling of the abbreviations as follows

5. We have corrected the spelling errors in this section and revised the punctuation as follows:

6. We have corrected “confidence level “ to confidence interval ( CI)

7. We have adjusted the spacing in the title of the forest plot figure.

8. We have corrected the spelling of “allover” to “overall” in the figure title.

9. We have revised the sentence “ When the individual studies were omitted, the pooled estimated prevalence……………….”, based the suggestion .

10. We agree that analyzing the disparities in visual impairment between Type 1 and Type 2 Diabetes would provide additional valuable insights. But the data is

11. We have corrected the spelling of “caus” to “cause” in the Causes of Visual Impairment section

[Discussion]

Can you look for similar reports in Western countries for comparison?

• What is the plausible explanation for the highest prevalence of visual impairment in Sudan?

References

Some of the reference lists did not have the complete list of information required for the referencing style you used. Please review and revise it accordingly.

Response:

• We acknowledge the importance of comparing our findings with similar reports from other regions. However, we encountered challenges in finding directly comparable studies. While we did identify one study, it specifically assessed visual impairment among patients with diabetic retinopathy, which differs from the broader focus of our study.

• We have reviewed and revised the reference list to ensure completeness and adherence to the required reference style.

Reviewer 2

[Major comments]:

You report that “the majority (of reviewed studies) were followed hospital-based cross-sectional study designs”. This could be a major limitation to the study, considering the difficulties to generalize hospital-based results to population level. This could also explain the particularly high prevalence of visual impairment in the diabetic patient samples.

There is no description of how visual acuity has been measured in each study, whether it was best-corrected visual acuity, and whether there were significant differences in the measurements between studies. This could be a second major limitation to the study.

There are also multiple grammatical errors and typos throughout the text. I highly recommend re-checking the manuscript thoroughly for these and consult an English language expert if needed.

Response: Thank you very much for your critical comment. We found your comments and concerns are very insightful and valuable.

• We acknowledge that hospital-based cross-sectional studies may limit the generalizability of our findings to the broader population. To address this concern, we have stated this as a limitation in the discussion section. Additionally, we clarify that almost all the reviewed studies followed an institutional-based cross-sectional design. However, one study did not specify its study design, despite being conducted in an institutional setting.

• Most of the included studies did not specify whether they reported best-corrected visual acuity or uncorrected visual acuity. However, almost all studies used the WHO classification of visual impairment. Since the WHO classification is based on the best-corrected visual acuity in the better-seeing eye, it is reasonable to assume that best-corrected visual acuity was the primary measure in these studies. However, we have now discussed this as a limitation in the Limitations section, acknowledging that differences in visual acuity measurement techniques may have introduced variability in the results.

• We have thoroughly reviewed the manuscript and corrected grammatical errors and typos.

[Minor comments]:

Consider removing the WHO classification of visual impairment from the Introduction as it is more properly included in the Materials and Methods.

Uniform the presentation of the data in Table 1 for better clarity.

Recent population-based studies have reported a significant decrease in visual impairment among diabetic patients (see references below). Considering the differing results in the current study, you should add more discussion about this difference.

Laatikainen et al. 2016 Improving visual prognosis of the diabetic patients during the past 30 years based on the data of the Finnish Register of Visual Impairment

Purola et al. 2022 Changes in Visual Impairment due to Diabetic Retinopathy During 1980–2019 Based on Nationwide Register Data

Response: Thank you for your helpful suggestions.

- As per your recom

---

## [Decision Letter · Decision Letter 1]

Dear Dr. Kassaw,

Thank you for submitting your manuscript to PLOS ONE. After careful consideration, we feel that it has merit but does not fully meet PLOS ONE’s publication criteria as it currently stands. Therefore, we invite you to submit a revised version of the manuscript that addresses the points raised during the review process.

We look forward to receiving your revised manuscript.

Kind regards,

Tebelay Dilnessa, MSc

Academic Editor

PLOS ONE

Additional Editor Comments:

 Still, it requires a maor revision throughly.

Reviewers' comments:

Reviewer's Responses to Questions

**Comments to the Author**

Reviewer #2: (No Response)

Reviewer #3: (No Response)

2. Is the manuscript technically sound, and do the data support the conclusions?

Reviewer #2: No

Reviewer #3: (No Response)

3. Has the statistical analysis been performed appropriately and rigorously?

Reviewer #2: N/A

Reviewer #3: (No Response)

4. Have the authors made all data underlying the findings in their manuscript fully available?

Reviewer #2: Yes

Reviewer #3: (No Response)

5. Is the manuscript presented in an intelligible fashion and written in standard English?

Reviewer #2: Yes

Reviewer #3: (No Response)

Reviewer #2: Thank you for addressing the concerns raised in the last review. Unfortunately, I do not feel that the responses regarding the lack of description of visual acuity measurements and the difficulties to generalize the data to population level are satisfactory, which would be critical considering that both limitations can lead to unacceptable bias for the analyses and conclusions of the study.

Reviewer #3: (No Response)

**Do you want your identity to be public for this peer review?** For information about this choice, including consent withdrawal, please see our Privacy Policy

Reviewer #2: No

Reviewer #3: No

---

## [Author Response · Author response to Decision Letter 2]

5 May 2025

Response to Reviewers’ Comments

Reviewer 2

[General comment]

“Thank you for addressing the concerns raised in the last review. Unfortunately, I do not feel that the responses regarding the lack of description of visual acuity measurements and the difficulties to generalize the data to population level are satisfactory, which would be critical considering that both limitations can lead to unacceptable bias for the analyses and conclusions of the study”.

Response: We sincerely thank the reviewer’s continued feedback and for highlighting these important concerns. We understand and acknowledge the critical importance of clearly describing visual acuity measurements and addressing generalizability to minimize potential biases.

We have revised the manuscript to provide a more detailed explanation of how visual acuity was measured in our study. These details have been added as supplementary material, S4. Specifically, we have now included:

• The method and chart used for measuring visual acuity (e.g., Snellen, ETDRS, or other).

• Whether measurements were taken with best-corrected or presenting visual acuity.

Regarding generalizability, we have added a discussion of these limitations in the revised manuscript.

[Title]:

What is the importance of “2010–2023”? … Better to mention it in the methodology part.

Response: We thank the reviewer for this helpful observation, and we have revised accordingly.

[Introduction]:

“Refine the language like: Although several primary studies have revealed that the burden of VI among DM patients is high, their results have revealed substantial variation regarding its prevalence in SSA.. not good sentence,…..and the like.

Moreover, the majority of these research were single-centered, had small sample sizes, and were institution based……………can’t be a reason to carry out your study. See it again”.

Response: We thank the reviewer for this valuable feedback. We agree the original introduction needed clearer language. We have revised the paragraph to improve structure and flow. We also strengthened the rationale for this review by highlighting the need for comprehensive synthesis due to methodological heterogeneity and the lack of pooled regional estimates, beyond just study design limitations.

[Method]:

• “I found this section very interesting. The authors clearly apply every step of the PRISMA guideline with in the main document and in the supplementary materials.

• You have used JBI method since your study is a prevalence.

Comments:

• Write the eligibility criteria in paragraph form

• Operationalize some outcome variables”

Response: We appreciate the reviewer’s positive feedback on the Methods section and are grateful for his/her constructive suggestions.

We have revised the eligibility criteria subsection to present the inclusion and exclusion criteria in paragraph format, as suggested by the reviewer.

In the revised manuscript, we have added operational definitions for key outcome variables, such as "vision impairment" and its different levels (e.g., mild, moderate, severe, blindness), based on the definitions used in the included studies. We have specified how these variables were measured and categorized within the context of our review.

[Result]:

Comments: I found it interesting, but refine the language, example

“A significant heterogeneity was seen in the primary studies that were incorporated into this systematic review and meta-analysis. Therefore, to look into the sources of heterogeneity, a sub- group analysis based on the country where the study was conducted was carried out.”……..re-write.

[Discussion]

Comments: • Generally shallow, fragmented ideas (flowless), lack of art of article writing, not supported by evidences………..make it somewhat deep and keep the flow and support with evidence. This part needs major revision.

Response: We sincerely appreciate the reviewer’s thoughtful and detailed critique of the Discussion section. We acknowledge that this section requires substantial revision to improve its depth, flow, and support with evidence. In response, we have substantially revised the entire Discussion section. We have taken the reviewer's comments seriously and have made significant changes to address the concerns.

Specific Comments and Responses:

Comments: “Some of the sentences in the first paragraph are not necessary, for example, 'The search was limited to articles published between 2010 and 2022, with a total of 26 articles published in the last thirteen years...already mentioned earlier”

Response: We agree that this sentence is redundant. We have removed it from the revised Discussion section.

---

## [Decision Letter · Decision Letter 2]

Dear Dr. Kassaw,

Thank you for submitting your manuscript to PLOS ONE. After careful consideration, we feel that it has merit but does not fully meet PLOS ONE’s publication criteria as it currently stands. Therefore, we invite you to submit a revised version of the manuscript that addresses the points raised during the review process.

We look forward to receiving your revised manuscript.

Kind regards,

Tebelay Dilnessa, MSc

Academic Editor

PLOS ONE

Journal Requirements:

Additional Editor Comments:

The paper was improved significantly, still it requires intensive revision. That is, it requires a through edition, revision and proofreading in terms of typographically, punctuation and grammatically.Avoid the year interval from the title, and you can include it somewhere for example in the inclusion/exclusion criteriaThe background of the abstract was no explanatory.The discussion should be supported by reasons for variation or similarity, not merely comparison. Additionally, the majority of the references/citations used in the discussion was not systematic review and meta-analysis and (SRMA). Therefore, it is batter you focus and use SRMA articles.Avoid the asterisks from list of referencesFollow the standard binomial nomenclature, italize journal name and the word ‘et al’

Reviewers' comments:

Reviewer's Responses to Questions

**Comments to the Author**

Reviewer #3: All comments have been addressed

2. Is the manuscript technically sound, and do the data support the conclusions?

Reviewer #3: Yes

3. Has the statistical analysis been performed appropriately and rigorously?

Reviewer #3: Yes

4. Have the authors made all data underlying the findings in their manuscript fully available?

Reviewer #3: Yes

5. Is the manuscript presented in an intelligible fashion and written in standard English?

Reviewer #3: Yes

Reviewer #3: The discussion section could benefit from a deeper exploration of the implications of the findings.

**Do you want your identity to be public for this peer review?** For information about this choice, including consent withdrawal, please see our Privacy Policy

Reviewer #3: No

---

## [Author Response · Author response to Decision Letter 3]

21 May 2025

Dear Academic Editor and Reviewers,

Thank you once again for your continued consideration of our manuscript, “Prevalence of Vision Impairment among Patients with Diabetes Mellitus in sub-Saharan Africa: A Systematic Review and Meta-Analysis from 2010–2023” (PONE-D-24-15939R1). We sincerely appreciate the opportunity to revise and resubmit our work for further evaluation.

We have carefully reviewed the additional comments and have made the necessary revisions to address all remaining concerns. We believe these changes have significantly strengthened the manuscript and that it now fully meets PLOS ONE’s publication criteria.

General Editorial Comments

1. Intensive proofreading needed (grammar, punctuation, typographical errors).

Response: We have carefully revised the entire manuscript for grammar, punctuation, and clarity.

2. Avoid the year interval from the title

Response: The title has been revised to remove the year interval. We now indicate the study period (2010–2023) under the Methods section (inclusion/exclusion criteria).

3. Abstract background not explanatory.

Response: The background section of the abstract has been revised to provide clearer context regarding the importance of the study and its public health implications.

4. Discussion lacks explanations for variations/similarities and overuses non-SRMA references.

Response: Thank you for your insightful comment. Following your suggestion, we have revised the discussion section to include explanatory reasoning for the regional variations and similarities in the prevalence of vision impairment among diabetic patients. While we encountered difficulty getting a sufficient number of regionally comparable similar SRMAs, we have incorporated large-scale nationwide studies to support our comparisons.

5. Avoid asterisks in references.

Response: All asterisks have been removed from the reference list.

6. Follow standard binomial nomenclature, italicize journal names and ‘et al.’:

Response: We have revised the references to follow standard binomial nomenclature and formatting, including italicizing journal names and ‘et al.’ as required.

We hope the revisions address your concerns and improve the manuscript's clarity and rigor. We appreciate the opportunity to revise and resubmit our work.

Kind regards,

Altaseb Beyene Kassaw

---

## [Editor Report · Decision Letter 3]

Prevalence of Vision Impairment among Patients with Diabetes Mellitus in sub-Saharan Africa: a Systematic Review and Meta-Analysis

PONE-D-24-15939R3

Dear Dr. Kassaw,

We’re pleased to inform you that your manuscript has been judged scientifically suitable for publication and will be formally accepted for publication once it meets all outstanding technical requirements.

Kind regards,

Tebelay Dilnessa, MSc

Academic Editor

PLOS ONE

Additional Editor Comments (optional):

Figure 1 was not visible enough, please make it clear during submissions of the proofread.
---

## [Editor Report · Acceptance letter]

PONE-D-24-15939R3

PLOS ONE

Dear Dr. Kassaw,

I'm pleased to inform you that your manuscript has been deemed suitable for publication in PLOS ONE. Congratulations! Your manuscript is now being handed over to our production team.

Kind regards,

on behalf of

Dr. Tebelay Dilnessa

Academic Editor

PLOS ONE